# Adaptive Hinge Balance Loss for Document-Level Relation Extraction

**Jize Wang**[1], **Xinyi Le**[1]*, **Xiaodi Peng**[2] and **Cailian Chen**[1]

[1]Department of Automation, Shanghai Jiao Tong University
[2]Inspur Genersoft Co., Ltd.

{jizewang2000, lexinyi, cailianchen}@sjtu.edu.cn
pengxd@inspur.com

## Abstract

Document-Level Relation Extraction aims at predicting relations between entities from multiple sentences. A common practice is to select multi-label classification thresholds to decide whether a relation exists between an entity pair. However, in the document-level task, most entity pairs do not express any relations, resulting in a highly imbalanced distribution between positive and negative classes. We argue that the imbalance problem affects threshold selection and may lead to incorrect "no-relation" predictions. In this paper, we propose to downweight the easy negatives by utilizing a distance between the classification threshold and the predicted score of each relation. Our novel Adaptive Hinge Balance Loss measures the difficulty of each relation class with the distance, putting more focus on hard, misclassified relations, i.e. the minority positive relations. Experiment results on Re-DocRED demonstrate the superiority of our approach over other balancing methods. Source codes are available at https://github.com/Jize-W/HingeABL.

## 1 Introduction

Document-Level Relation Extraction (RE) plays an important role in NLP applications such as knowledge graph construction. It aims at predicting relations between entities from multiple sentences. As illustrated in Figure 1a, an entity pair may have zero, one, or multiple relations, so document-level RE is a multi-label classification task. To solve this, a common practice is to adaptively select thresholds for multi-label classification (Zhou et al., 2021). For a correct prediction, the confidence scores of existent relations should be higher than the threshold, and conversely, those of non-existent relations should be lower.

However, there is a significant imbalance problem between positive and negative classes in document-level RE. The number of entity pairs

*Corresponding author.

---

**Text:** Ross Patterson Alger (August 20, 1920 - January 16, 1992) was a politician in the *Canadian* province of Alberta, ... After the war, he received an MBA from the *University of Toronto*. He settled in Calgary and started a career in accounting ...

---

**Subject:** *University of Toronto*
**Object:** *Canadian*
**Relation:** country, located in

(a) A sample document in Re-DocRED dataset.

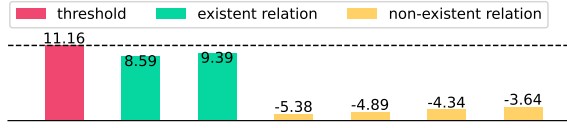

(b) False negative prediction with correct label ranking.

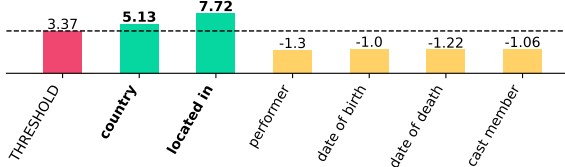

(c) Correct prediction after utilizing adaptive hinge balance loss.

Figure 1: **Illustration on multi-label classification in document-level relation extraction.** (a) There are two relations existing between *University of Toronto* and *Canadian*. (b) The entity pair in (a) is incorrectly predicted as "no-relation". Scores of existent relations (*country*, *located in*) are lower than the threshold (11.16), but significantly higher than all non-existent relations. (c) After utilizing adaptive hinge balance loss, the threshold is reduced to an appropriate value.

increases quadratically with the number of entities. Thus, compared with the sentence-level counterpart, there are far more entity pairs to be classified in document-level RE, and most entity pairs express no relation. For example, in the document-level RE dataset, Re-DocRED (Tan et al., 2022b), 94% of the entity pairs express no relation.

The issue of class imbalance may lead to more incorrect "no-relation" predictions. In our paper, we mainly consider the "positive/negative imbalance", rather than the imbalance between different

types of "positive relations". The positive/negative imbalance tends to drag the threshold towards the large class, that is, the "no-relation" class. We discover that **78.2% of incorrect "no-relation" predictions have correct label ranking**, but the confidence score is lower than the threshold, which is shown in Figure 1b . In other words, the model has enough confidence in the existent relations, but it makes a "no-relation" prediction due to the unnecessarily high threshold.

Based on this intuitive finding, we aim to address this "incorrect predictions with correct label ranking" phenomenon to improve the accuracy. We believe that overtraining on well-classified non-existent relations may lead to unnecessarily high thresholds. Therefore, we propose to adaptively select thresholds, and then down-weight the relations that are far from the decision boundary using Hinge Weighting. Our contributions are three-fold:

- We design a general pipeline termed Separate Adaptive Thresholding, to adaptively select thresholds for multi-label classification.

- We propose a novel Adaptive Hinge Balance Loss, tackling the imbalance problem of positive and negative classes in document-level RE.

- Among all the existing balancing methods, our method achieves the highest F1 score on the common dataset Re-DocRED.

## 2 Preliminary

The task of document-level relation extraction is concerned with the prediction of relation types between subject and object entities in a given document. We will first introduce the formulation of this task, and then discuss the commonly used ATL method.

### 2.1 Problem Formulation

Given a document $D$ that contains a set of entities $\{e_i\}_{i=1}^n$, the task of document-level relation extraction is to predict the relation types between the entity pairs $(e_s, e_o)_{s,o \in \{1,...,n\}, s \neq o}$, where $e_s$ and $e_o$ represent the subject entity and the object entity, respectively. The set of relations is defined as $\mathcal{R} \cup \{NA\}$, where $\mathcal{R}$ is a set of pre-defined relations and NA stands for no relation between a pair of entities.

With the document $D$ and an entity pair $(e_s, e_o)$ contained in it, we can get the representation of the subject and object entity through:

$$[\mathbf{z}_s, \mathbf{z}_o] = Rep(D, e_s, e_o), \qquad (1)$$

where $\mathbf{z}_s$ and $\mathbf{z}_o$ are the representation of the subject and object entity. $Rep$ is a representation module.

The score of relation $r$ is defined as $s_r$, which can be computed via the subject and object entity representation using a bilinear classifier:

$$s_r = \mathbf{z}_s^T \mathbf{W}_r \mathbf{z}_o + b_r, \qquad (2)$$

where $\mathbf{W}_r \in \mathbb{R}^{d \times d}, b_r \in \mathbb{R}$ are model parameters.

### 2.2 Adaptive Thresholding Loss

Adaptive Thresholding Loss (ATL) (Zhou et al., 2021) is the most widely used loss function in transformer-based document-level relation extraction methods. It enables the model to choose multi-label classification thresholds, thereby achieving superior results when compared to the global threshold of BCE loss (Bengio et al., 2013).

In ATL, the labels of entity pair $T = (e_s, e_o)$ are divided into two subsets: positive classes $\mathcal{P}_T$ and negative classes $\mathcal{N}_T$, where $\mathcal{P}_T \subseteq \mathcal{R}$ denotes the relations that exist between $T$, and $\mathcal{N}_T \subseteq \mathcal{R}$ denotes the relations that do not exist between the entities. ATL introduces an additional threshold class TH. If an entity pair is correctly classified, the scores of $\mathcal{P}_T$ should be higher than TH while those of $\mathcal{N}_T$ should be lower. ATL comprises of two parts:

$$\mathcal{L}_1 = -\sum_{r \in \mathcal{P}_T} \log \left( \frac{\exp(s_r)}{\sum_{r' \in \mathcal{P}_T \cup \{TH\}} \exp(s_{r'})} \right),$$
$$(3)$$

$$\mathcal{L}_2 = -\log \left( \frac{\exp(s_{TH})}{\sum_{r' \in \mathcal{N}_T \cup \{TH\}} \exp(s_{r'})} \right), \qquad (4)$$

$$\mathcal{L}_{ATL} = \mathcal{L}_1 + \mathcal{L}_2. \qquad (5)$$

### 2.3 An Empirical Analysis of ATL

A preliminary analysis is conducted to investigate the cause of classification error in ATL, as shown in Table 1. All false predictions can be categorized into three patterns: FP, FN_CRK, and FN_IRK, which are illustrated in Figure 2. In particular, FN_CRK is the most dominant source of errors, which accounts for 78.2% of all false negative predictions.

We notice that the number of relations in $\mathcal{N}_T$ is significantly larger than that in $\mathcal{P}_T$, and therefore

| # FP | # FN_CRK | # FN_IRK |
|------|----------|----------|
| 2859 | **3192** | 889 |

Table 1: Number of three false patterns of ATL's predictions on Re-DocRED. Three patterns are illustrated in Figure 2.

$\mathcal{L}_2$ has a much greater impact on the overall loss than $\mathcal{L}_1$ (see equations (3) and (4)). Due to the dominance of $\mathcal{L}_2$, it can be rewritten as the following form:

$$\mathcal{L}_2 = -\log\left(\frac{1}{1 + \sum_{r' \in \mathcal{N}_T} \exp(s_{r'} - s_{\text{TH}})}\right).$$
(6)

$\mathcal{L}_2 \to 0$ when $s_{r'} - s_{\text{TH}} \to -\infty$, which means $s_{\text{TH}} \gg s_{r'}$. This suggests ATL learns a threshold $s_{\text{TH}}$ well above the candidate score, which leads to an increase in the number of FN_CRK predictions.

## 3 Adaptive Hinge Balance Loss

Based on the analysis above, we aim to maximize the distance between the decision boundary $s_{\text{TH}}$ and the sample point $s_r, r \in \mathcal{R}$ while simultaneously down-weighting the classes distant from the boundary. To this end, we propose our Adaptive Hinge Balance Loss.

### 3.1 Separate Adaptive Thresholding

An ideal loss should maximize the distance from the decision boundary to the sample point. Moreover, in the loss formulation, each relation class should be independent of the others to enable individual weighing of each class. Therefore, we propose the Separate Adaptive Thresholding (SAT), which is formulated as:

$$\mathcal{L} = -\sum_{r \in \mathcal{R}} \log(\sigma(-d_r)),$$
(7)

$$d_r = \begin{cases} s_r - s_{\text{TH}} & r \in \mathcal{P}_T \\ s_{\text{TH}} - s_r & r \in \mathcal{N}_T \end{cases}$$
(8)

where $\sigma$ is the sigmoid function, i.e. $\sigma(x) = \frac{1}{1+e^x}$. We define $s_{\text{TH}}$ as the decision boundary. Then $d_r$ is the distance from the decision boundary to the score of relation $r \in \mathcal{R}$. $d_r > 0$ if a relation is correctly classified. $\mathcal{L} \to 0$ when $d_r \to \infty$.

The loss pushes $d_r$ to be as large as possible. The score of each relation is compared with the threshold separately. Thus we can assign different weights to different relations.

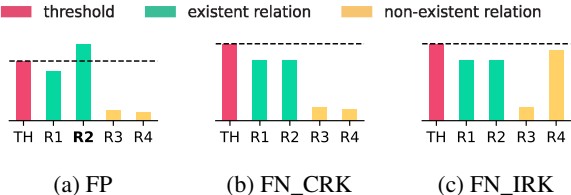

(a) FP  (b) FN_CRK  (c) FN_IRK

Figure 2: **Three false prediction patterns.** (a) **FP** (False Positive): The entity pair is recognized as related, but not all relations are accurately recognized. (b) **FN_CRK** (False Negative with Correct label RanKing): The entity pair is recognized as "no-relation", and all the existent relations have higher confidence scores than non-existent relations. (c) **FN_IRK** (False Negative with Incorrect label RanKing): The entity pair is recognized as "no-relation", and not all existent relations have higher confidence scores than non-existent relations.

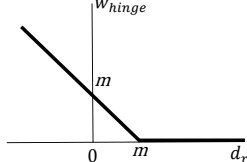

Figure 3: Hinge weighting.

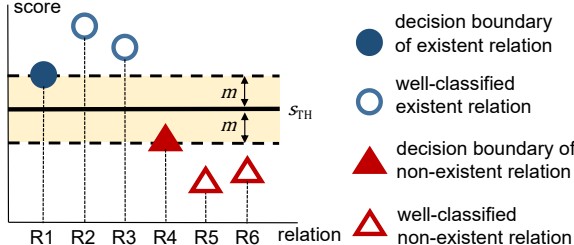

Figure 4: **The decision boundary and margin of hinge weighting.** Positive relations whose scores are higher than $s_{\text{TH}} + m$ and negative relations whose scores are lower than $s_{\text{TH}} - m$ will not be punished.

### 3.2 Hinge Weighting

To down-weight the easy and well-classified relations, i.e. the majority negative relations, we propose Hinge Weighting inspired by hinge loss. (Hearst et al., 1998)

Our Hinge Weighting is shown in Figure 3. It is formulated as:

$$w_r = \max(0, m - d_r), r \in \mathcal{R},$$
(9)

where $m$ is a constant. When the distance $d_r$ is larger than $m$, the relation is not penalized. Otherwise, it is penalized linearly with $d_r$. Essentially, Hinge Weighting implies that we should avoid focusing on the relationship with large $d_r$. $2m$ is the maximum margin between positive and negative classes, which is illustrated in Figure 4.

Note that our weighting mechanism downweights the well classified samples to zero. Since

| Model | F1 | F1 with HingeABL | Ign_F1 | Ign_F1 with HingeABL |
|---|---|---|---|---|
| ATLOP (Zhou et al., 2021) | 77.56 | **79.79** (+2.23) | 76.82 | **78.82** (+2.00) |
| DocuNet (Zhang et al., 2021) | 77.87 | **79.43** (+1.56) | 77.26 | **78.39** (+1.13) |
| KD-DocRE (Tan et al., 2022a) | 78.28 | **79.34** (+1.06) | 77.60 | **78.26** (+0.66) |

Table 2: Performance of adaptive hinge balance loss with different backbones.

the majority of well classified classes are negative, HingeABL achieves the effect of down-weighting the majority negative relations.

### 3.3 Loss Definition

Combining separate adaptive thresholding and hinge weighting, we obtain the adaptive hinge balance loss (HingeABL):

$$\mathcal{L} = -\sum_{r \in \mathcal{R}} \frac{w_r}{\sum_{r' \in \mathcal{R}} w_{r'}} \log(\sigma(-d_r)), \quad (10)$$

where $\sigma$ is the sigmoid function and $w_r$ is formulated as Equation (9). The hinge weights are normalized among all relations. Our adaptive hinge balance loss simultaneously maximizes the distance between the decision boundary and the sample point and down-weights easy classes that are far from it. This helps prevent over-fitting on well-classified relations.

## 4 Experiments

### 4.1 Setup

We conduct experiments on Re-DocRED (Tan et al., 2022b), the largest and well-labeled dataset for document-level RE. We use F1 and Ign_F1 as the metrics. Ign_F1 is measured by removing the relations existing in the training set from the dev/test sets. More details about statistics and implementation are provided in Appendix A and B. Note that we use micro F1 here in order to maintain consistency with previous methods. However, macro F1 is more suitable to illustrate whether the proposed method can perform better on minority classes. Results evaluated under macro F1 are provided in Appendix C.

### 4.2 Results

**Different balancing methods.** To compare different balancing methods, we use ATLOP (Zhou et al., 2021) as the representation module and BERT$_{base}$ (Devlin et al., 2019) as the encoder of it. We also compare our method with three other approaches: Balanced Softmax (Zhang et al., 2021), AML (Adaptive Margin Loss) (Wei and Li, 2022), and AFL (Adaptive Focal Loss) (Tan et al., 2022a).

| Loss Function | F1 | Ign_F1 |
|---|---|---|
| ATL (Zhou et al., 2021) | 73.29 | 72.46 |
| Balanced Softmax (Zhang et al., 2021) | 73.68 | 72.85 |
| AML (Wei and Li, 2022) | 72.60 | 71.78 |
| AFL (Tan et al., 2022a) | 74.15 | 73.20 |
| SAT | 73.46 | 72.61 |
| MeanSAT | 74.68 | 72.90 |
| HingeABL | **75.15** | **73.84** |

Table 3: Comparison with other balancing methods.

Both AML and HingeABL are margin-based loss functions.

To illustrate the effectiveness of hinge weighting, we implement an alternative weighted loss called MeanSAT by weighting positive and negative classes of SAT by the inverse of their number. Its formulation is in Appendix D.

The results are shown in Table 3. HingeABL achieves the highest F1 and Ign_F1 of 75.15 and 73.84 among all balancing methods. We observe a substantial increase in performance by implementing two weighting methods on the SAT. Our experiments indicate that hinge weighting surpasses constant weighting with MeanSAT, which demonstrates the superiority of HingeABL.

Besides, we compare the two margin-based loss functions, AML and HingeABL, through mathematical analysis, provided in Appendix E. We find that AML penalizes the misclassified samples linearly with the distance, while HingeABL penalizes the misclassified samples nonlinearly with the distance. The nonlinear function is strictly convex, which benefits optimization.

**Different document-level RE models.** To test the generality of our approach, we select three commonly used transformer-based methods for document-level relation extraction and replace their loss functions with our adaptive hinge balance loss. Among the three original base methods, ATLOP employs ATL loss, DocuNet employs Balanced Softmax loss, and KD-DocRE employs AFL loss. Both Balanced Softmax and AFL are the improvements of ATL. All methods use RoBERTa$_{large}$ (Zhuang et al., 2021) as their encoder. Table 2 shows the results, all of which demonstrate consis-

| Loss Function | F/(T+F) | FN/F | FN_CRK /FN |
|---|---|---|---|
| ATL | 3.59% | 58.80% | 78.22% |
| Balanced Softmax | 4.84% | 51.16% | 79.95% |
| AML | 3.54% | 65.30% | 57.14% |
| AFL | 3.59% | 52.95% | 75.98% |
| HingeABL | **3.49%** | **51.11%** | **43.84%** |

Table 4: Statistics of prediction patterns for different loss functions.

tent performance gains with the use of HingeABL. This affirms the generalizability of our approach. Note that HingeABL's improvement seems to be less significant when the base method is more powerful. This is a natural result because better base methods employ better loss functions. Replacing a better loss function with HingeABL results in a smaller improvement.

**Prediction statistics.** To verify whether our model solves the problem of high thresholds, we count the number of prediction patterns from Figure 2 and present the results in Table 4. Our analysis reveals that the proportion of FN and FN_CRK has decreased, indicating that the issue has been resolved. An example of prediction results before and after applying HingeABL is shown in Figure 1b and 1c. While one might assume that lowering the threshold would lead to more false positive predictions, we observe that the total proportion of false predictions actually decreases. This suggests that HingeABL achieves a good balance in its threshold selection.

## 5 Conclusion

We propose a novel Adaptive Hinge Balance Loss for document-level relation extraction to tackle the imbalance problem of positive and negative classes. Experimental results show our approach outperforms existing methods. Since our loss is model-independent, it has potential applicability to other multi-label classification scenarios.

## Limitations

Compared with classifying an entity pair known with relation, accurately determining whether a relation exists between an entity pair is a more challenging task. Despite attempts to improve accuracy through better thresholding methods, the results are still far from ideal.

## Acknowledgements

This work was supported by the National Key Research and Development Program of China (2021YFB1716000) and the National Natural Science Foundation of China (No.62176152).

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

## A  Re-DocRED Statistics

Re-DocRED is a more reliable benchmark in document-level relation extraction. It is a revised version of DocRED (Yao et al., 2019), whose annotations are pointed out to be incomplete by recent works ((Huang et al., 2022; Tan et al., 2022a)). Re-DocRED contains 96 relations. Each document has an average of 391 entity pairs, among which 94% contains no relation. The detailed statistics are shown in Table 5 and Table 6.

| # Relations | 96 |
|---|---|
| Avg. # Words | 198.4 |
| Avg. # Entities | 19.4 |
| Avg. # Entity Pairs | 391.0 |
| NA | 94% |

Table 5: Statistics on the whole set of Re-DocRED.

| | Train | Dev | Test |
|---|---|---|---|
| # Documents | 3053 | 500 | 500 |
| Avg. # Entities | 19.4 | 19.4 | 19.6 |
| Avg. # Triples | 28.1 | 34.6 | 34.9 |
| Avg. # Sentences | 7.9 | 8.2 | 7.9 |

Table 6: Statistics on different train/dev/test dataset of Re-DocRED.

## B  Implementation Details

All experiments are implemented based on Hugging Face's Transformers (Wolf et al., 2020). In the experiment of comparing different balancing methods, we use BERT$_{base}$ (Devlin et al., 2019) as the encoder of ATLOP. In the experiment of comparing different RE models, we use RoBERTa$_{large}$ (Devlin et al., 2019) as the encoder of these RE models, for the sake of comparison on the benchmark.

We select the margin of HingeABL as $m = 5$ when conducting experiments. We use mixed-precision training (Micikevicius et al., 2018) based on the PyTorch amp library[1]. The models are optimized with AdamW (Loshchilov and Hutter, 2019) with a linear warmup (Goyal et al., 2017) for the first 6% steps followed by a linear decay to 0. The learning rate is 5e-5 for models with BERT as the encoder and 3e-5 for models with RoBERTa as the encoder. The train batch size is 4 and the test batch size is 8. We train 30 epochs for each model. For each experiment, we run 5 different seeds (1, 5, 42, 66, 233) and report the average score. All models are trained with 1 Tesla A800 GPU.

## C  Comparison results under macro F1

The comparison results among different balancing methods under macro F1 and macro Ign_F1 are shown in Table 7. Our proposed HingeABL still achieves the highest score under macro F1.

## D  Formulation of MeanSAT

MeanSAT weights positive and negative classes of SAT by the inverse of their number. It is formulated

---

[1]https://pytorch.org/docs/stable/amp.html

| Loss Function | F1 | Ign_F1 |
|---|---|---|
| ATL (Zhou et al., 2021) | 59.39 | 56.57 |
| Balanced Softmax (Zhang et al., 2021) | 60.67 | 57.89 |
| AML (Wei and Li, 2022) | 58.65 | 55.81 |
| AFL (Tan et al., 2022a) | 61.48 | 58.66 |
| SAT | 60.23 | 57.41 |
| MeanSAT | 63.34 | 60.91 |
| HingeABL | **64.13** | **61.34** |

Table 7: Comparison with other balancing methods under macro F1 and macro Ign_F1.

as:

$$\mathcal{L} = -\frac{1}{N_p}\sum_{r\in\mathcal{P}}\log(\sigma(-d_r)) - \frac{1}{N_n}\sum_{r\in\mathcal{N}}\log(\sigma(-d_r)), \quad (11)$$

where $N_p$ and $N_n$ are the number of positive and negative classes for the entity pair.

## E  Mathematical analysis of Adaptive Margin Loss and HingeABL

In addition to experiments, we also compare the two margin-based losses, Adaptive Margin Loss (AML) and HingeABL, from a mathematical analysis perspective.

**Analysis 1: For a sample that is not well classified, the Adaptive Margin Loss is a linear function with respect to the distance.**

The Adaptive Margin Loss is defined as:

$$\mathcal{L} = \sum_{r\in\mathcal{R}}\max(0, m - d_r). \quad (12)$$

For class $r$, $\mathcal{L}_r = \max(0, m - d_r)$.
For a well-classified sample, $d_r \geq m$, $\mathcal{L}_r = 0$.
For a sample that is not well classified, $d_r < m$, $\mathcal{L}_r = m - d_r$.

In the second condition, we denote $c_r = -d_r > -m$. It measures the distance between a sample that is not well classified to the decision boundary. Note that we call $c_r$ "distance" here, but it is not necessarily greater than zero. The smaller $c_r$ is, the better the sample is classified. This means we should give a larger punishment to a larger $c_r$. Then we have:

$$\mathcal{L}_r = m + c_r, \quad (13)$$

$$\frac{\partial\mathcal{L}_r}{\partial c_r} = 1. \quad (14)$$

This means Adaptive Margin Loss penalizes the samples that are not well classified linearly with the distance. (Note: A sample that is not well classified means $c_r = -d_r > -m$. A sample that is misclassified means $c_r = -d_r > 0$.)

**Analysis 2: For a sample that is not well classified, HingeABL is a strictly convex function with respect to the distance.**

HingeABL is defined as:

$$\mathcal{L} = -\sum_{r\in\mathcal{R}}\frac{w_r}{\sum_{r'\in\mathcal{R}}w_{r'}}\log(\sigma(-d_r)) \quad (15)$$

$$= -\sum_{r\in\mathcal{R}}\frac{\max(0, m - d_r)}{\sum_{r'\in\mathcal{R}}\max(0, m - d_{r'})}\log\left(\frac{1}{1 + e^{-d_r}}\right). \quad (16)$$

The denominator $\sum_{r'\in\mathcal{R}}\max(0, m - d_{r'})$ is a normalization factor, which we discard for ease of analysis.

For class $r$, if a sample is well classified, $d_r \geq m$, $\mathcal{L}_r = 0$.

If a sample is not well classified, $d_r < m$,

$$\mathcal{L}_r = -(m - d_r)\log\left(\frac{1}{1 + e^{-d_r}}\right) \quad (17)$$

$$= -(m + c_r)\log\left(\frac{1}{1 + e^{c_r}}\right), \quad (18)$$

$$\frac{\partial\mathcal{L}_r}{\partial c_r} = -\log\left(\frac{1}{1 + e^{c_r}}\right) + \frac{m + c_r}{e^{-c_r} + 1}, \quad (19)$$

$$\frac{\partial^2\mathcal{L}_r}{\partial c_r^2} = \frac{e^{c_r}}{1 + e^{c_r}} + \frac{e^{-c_r} + 1 + e^{-c_r}(m + c_r)}{(e^{-c_r} + 1)^2} > 0. \quad (20)$$

This means HingeABL penalizes the samples that are not well classified nonlinearly with the distance. The nonlinear function is strictly convex.

**Analysis 3: Comparision between the Adaptive Margin Loss and HingeABL.**

1. Similarities.

Both the Adaptive Margin Loss and HingeABL are margin-based loss functions. They do not punish a prediction if it is correct and "good enough" (rather than "perfect"), which is a form of regularization to prevent overfitting.

2. Differences.

For the wrong prediction part, they both give a penalty according to the distance $c_r$. The Adaptive Margin Loss gives a linear penalty, while HingeABL gives a strictly convex penalty. Compared to a linear penalty, a strictly convex penalty has mainly two advantages: 1. When $c_r$ is larger, HingeABL gives a larger penalty than the Adaptive Margin Loss. 2. Compared to linear functions, the nature of strictly convex functions makes the optimization more stable and more likely to converge to a globally optimal solution.