# OpenReview forum: "Adaptive Hinge Balance Loss for Document-Level Relation Extraction"
_EMNLP/2023/Conference — EMNLP 2023 Findings_

### Official Review · Reviewer_cr9k · 2023-08-01

**Typos Grammar Style And Presentation Improvements:** line 39, "squarely", do you mean "qua…
**Soundness:** 3

**Excitement:**

3: Ambivalent: It has merits (e.g., it reports state-of-the-art results, the idea is nice), but there are key weaknesses (e.g., it describes incremental work), and it can significantly benefit from another round of revision. However, I won't object to accepting it if my co-reviewers champion it.

**Missing References:**

Wei, Y., & Li, Q. (2022, August). SagDRE: Sequence-Aware Graph-Based Document-Level Relation Extraction with Adaptive Margin Loss. In Proceedings of the 28th ACM SIGKDD Conference on Knowledge Discovery and Data Mining (pp. 2000-2008).

**Paper Topic And Main Contributions:**

This work presents an adaptive Hinge balance loss for document-level relation extractions, addressing the critical issue of class imbalance in imbalanced relation types. The proposed method combines hinge loss and cross-entropy loss and is evaluated on real-world large-scale data using the Re-DocRED dataset, demonstrating its effectiveness.

**Questions For The Authors:**

The method did not seem to address the class imbalance issues. The imbalance issue means the number of samples for each class is not uniform, and some relations have very few samples. A common solution from the loss function perspective is to put higher weights on minority classes (a higher penalty for misclassifying a sample in a minority class). The proposed weighting mechanism doesn't seem to favor minority relations.

The evaluation metric is also not proper if the paper aims to address the imbalance issue. The F1 used by baseline methods is micro F1, giving equal importance to each sample regardless of the class it belongs to. So in the final score, the performance of the minority classes is hidden, and the performance of the majority classes is amplified. Macro F1 should be used instead to better illustrate if the proposed method can perform better on minority classes.

If the authors used the term "imbalance" to mean other issues, please clearly define it in the paper.

**Reasons To Accept:**

The research tackles an interesting and challenging problem concerning class imbalance in document-level relation extraction, which is a critical issue.

The adaptive Hinge loss is well-motivated and thoughtfully implemented through a combination of hinge loss and cross-entropy loss.

The evaluation utilizes real-world large-scale data, enhancing the reliability and practicality of the results.

**Reasons To Reject:**

The paper lacks a comparison with state-of-the-art models, particularly the one employing margin-based losses for document-level relation extraction [1]. Including such comparisons is crucial to understanding the performance of the proposed method relative to the existing techniques.

[1] Wei, Y., & Li, Q. (2022, August). SagDRE: Sequence-Aware Graph-Based Document-Level Relation Extraction with Adaptive Margin Loss. In Proceedings of the 28th ACM SIGKDD Conference on Knowledge Discovery and Data Mining (pp. 2000-2008).

The method did not address the class imbalance issues

**Reproducibility:**

4: Could mostly reproduce the results, but there may be some variation because of sample variance or minor variations in their interpretation of the protocol or method.

**Reviewer Confidence:**

5: Positive that my evaluation is correct. I read the paper very carefully and I am very familiar with related work.

---

> ### Author Rebuttal · Authors · 2023-08-29
>
> **Q1: The class imbalance issues, and the "imbalance" definition.**
>
> In our paper, we mainly consider the imbalance problems of the “positive/negative imbalance”, and ignore the imbalance between different types of “positive relations”. We find that, in our document-level task, the negative class is significantly larger than the positive class, which has a larger impact on the objective function.
> In the revised paper, we will clarify the "imbalance" into “positive/negative imbalance”.
>
> **Q2: Weighting mechanism to favor minority relations.**
>
> Refer to Q1, because we mainly tackle the imbalance problems of the “positive/negative imbalance”, “minority relations” here mean “positive relations”. Our weighting mechanism uses HingeABL to down-weight the majority negative class, thus favoring the “minority positive relations”.
>
> (Specifically, our weighting mechanism down-weights the well classified samples to zero. Since the majority of well classified classes are negative, HingeABL achieves the effect of down-weighting the majority negative class.)
>
> **Q3: Comparison with SagDRE.**
>
> Here we compare our method with Adaptive Margin Loss proposed by SagDRE from both the experiment results and the mathematical analysis.
> In our setting (ATLOP-BERT-base as the representation module, details in 4.2 “Different balancing methods” in our paper), our proposed HingeABL **still achieves the highest score** among all balancing methods. The revised comparison table is as follows. (Note: Evaluated under Micro F1. Scores under Macro F1 are shown in Q4.)
>
> |Loss Function|Micro F1|Micro Ign_F1|
> |---|---|---|
> |ATL|73.29|72.46|
> |Balanced Softmax|73.68|72.85|
> |AFL|74.15|73.20|
> |Adaptive Margin Loss (SagDRE) [1]|72.60|71.78|
> |SAT|73.46|72.61|
> |MeanSAT|74.68|72.90|
> |**HingeABL (Ours)**|**75.15**|**73.84**|
>
> Both HingeABL and Adaptive Margin Loss are margin-based. Adaptive Margin Loss penalizes the misclassified samples linearly with the distance, while HingeABL in our paper penalizes the misclassified samples **nonlinearly** with the distance, using logsoftmax function. We claim that a suitable nonlinear penalty benefits the “imbalance reduction”. The mathematical analysis is as follows:
>
> **Analysis 1: For a misclassified sample, the Adaptive Margin Loss is a linear function with respect to the distance.**
>
> The Adaptive Margin Loss is defined as:
> $$
> \mathcal{L}=\sum_{1\leq i \leq C}\text{max}(0,\alpha-t_i(z_i-z_s))
> $$
> where $t_i=1$ if $R_i\in\mathcal{P}$;  $t_i=-1$ if $R_i\in\mathcal{N}$.
>
> For class $i$,
> $\mathcal{L}_i=max(0,\alpha-t_i(z_i-z_s))$.
>
> For a well classified sample, $|z_i-z_s|\geq \alpha$,
> $\mathcal{L}_i=0$.
>
> Otherwise, $|z_i-z_s|< \alpha$, $\mathcal{L}_i=\alpha-t_i(z_i-z_s)$.
>
> In the second condition, we denote
> $d_i=-t_i(z_i-z_s)>0$. It measures the distance between a misclassified sample to the decision boundary. Then we have:
> $$\mathcal{L}_i=\alpha+d_i$$
> $$
> \frac{\partial \mathcal{L}_i}{\partial d_i}=1
> $$
>
> This means Adaptive Margin Loss penalizes the misclassified samples linearly with the distance.
>
> **Analysis 2: For a misclassified sample, HingeABL is a strictly convex function with respect to the distance.**
>
> For ease of comparison, we use the same set of symbols as above for HingeABL (Equation(10) in our paper):
> $$
> \mathcal{L}=-\sum_{1\leq i \leq C}w_i\text{log}(\sigma(-t_i(z_i-z_s))$$
> $$=-\sum_{1\leq i \leq C}\frac{max(0,\alpha-t_i(z_i-z_s))}{\sum_{1\leq i \leq C}max(0,\alpha-t_i(z_i-z_s))}\text{log}\left(\frac{1}{1+e^{-t_i(z_i-z_s)}}\right)
> $$
>
> The denominator $\sum_{1\leq i \leq C}max(0,\alpha-t_i(z_i-z_s))$ is a normalization factor, which we discard for ease of analysis.
>
> For class i, if a sample is well classified, $|z_i-z_s|\geq \alpha$, $\mathcal{L}_i=0$.
>
> Otherwise, $|z_i-z_s|< \alpha$,
> $$\mathcal{L}_i=-(\alpha-t_i(z_i-z_s))\text{log}\left(\frac{1}{1+e^{-t_i(z_i-z_s)}}\right)
> =-(\alpha+d_i)\text{log}\left(\frac{1}{1+e^{d_i}}\right)$$
>
> $$
> \frac{\partial \mathcal{L}_i}{\partial d_i} = -\text{log}\left(\frac{1}{1+e^{d_i}}\right)+\frac{\alpha+d_i}{e^{-d_i}+1}
> $$
>
> $$
> \frac{\partial^2\mathcal{L}_i}{\partial d_i^2}=\frac{e^{d_i}}{1+e^{d_i}}+\frac{e^{-{d_i}}+1+e^{-{d_i}}(\alpha+d_i)}{(e^{-d_i}+1)^2}>0
> $$
>
> This means HingeABL penalizes the misclassified samples nonlinearly with the distance. The nonlinear function is strictly convex.
>
> **Analysis 3: Comparision between the Adaptive Margin Loss and HingeABL.**
>
> 1.Similarities.
>
> Both the Adaptive Margin Loss and HingeABL are margin-based loss functions. They do not punish a prediction if it is correct and "good enough" (rather than "perfect"), which is a form of regularization to prevent overfitting.
>
> 2.Differences.
>
> For the wrong prediction part, they both give penalty according to the distance $d_i$. The Adaptive Margin Loss gives a linear penalty, while HingeABL gives a strictly convex penalty. Compared to a linear penalty, a strictly convex penalty has mainly two advantages:
>
> (1) When $d_i$ is larger, HingeABL gives a larger penalty than the Adaptive Margin Loss.
>
> (2) Compared to linear functions, the nature of strictly convex functions makes the optimization more stable and more likely to converge to a globally optimal solution.
>
> We will add the comparison of Adaptive Margin Loss in SagDRE to our paper. We believe that this kind of margin-based loss functions worths exploring in depth for more complex tasks.
>
> **Q4: The evaluation metric Macro F1.**
>
> Our proposed HingeABL **still achieves the highest score under Macro F1**. The results are shown in the table below.
>
> |Loss Function|Macro F1|Macro Ign_F1|
> |---|---|---|
> |ATL|59.39|56.57|
> |Balanced Softmax|60.67|57.89|
> |AFL|61.48|58.66|
> |Adaptive Margin Loss (SagDRE) [1]|58.65|55.81|
> |SAT|60.23|57.41|
> |MeanSAT|63.34|60.91|
> |**HingeABL (Ours)**|**64.13**|**61.34**|
>
> [1] Wei, Y., & Li, Q. (2022, August). SagDRE: Sequence-Aware Graph-Based Document-Level Relation Extraction with Adaptive Margin Loss. In Proceedings of the 28th ACM SIGKDD Conference on Knowledge Discovery and Data Mining (pp. 2000-2008).

---

### Official Review · Reviewer_zSKD · 2023-08-02

**Soundness:** 4

**Excitement:**

3: Ambivalent: It has merits (e.g., it reports state-of-the-art results, the idea is nice), but there are key weaknesses (e.g., it describes incremental work), and it can significantly benefit from another round of revision. However, I won't object to accepting it if my co-reviewers champion it.

**Paper Topic And Main Contributions:**

This paper proposes a novel approach called the Adaptive Hinge Balance Loss (HingeABL) for document-level relation extraction. They build upon the empirical observation that the previous method (ATL) suffers from errors primarily due to an inappropriate threshold classifying positive/negative classes. This issue arises from the class imbalance problem, where the negative class is significantly larger than the positive class, which has a larger impact on the objective function. To tackle this problem, the authors introduce hinge weighting to mitigate the impact of the dominant negative relations. By incorporating this hinge weighting mechanism, the impact of the negative class would be close to zero, addressing the threshold problem. Experimental results demonstrate that HingeABL significantly improves the F1 score for document-level relation extraction on three fundamental methods, and it successfully alleviates the threshold-related challenges.

**Questions For The Authors:**

According to Table 2, HingeABL's improvement appears to be less significant when the base model is more powerful. This raises the question of whether a strong base method could potentially alleviate the threshold problem. Do you have further experiments to provide additional statistics on the FN-CRK in these three methods?

**Reasons To Accept:**

1. The proposed method is strongly motivated
2. The paper exhibits a well-organized structure, addressing the problem at hand, identifying its root causes, presenting an effective solution, and supporting it with experimental evidence to demonstrate its efficacy.
3. HingeABL consistently demonstrates improvement over three methods.

**Reasons To Reject:**

None.

**Reproducibility:**

4: Could mostly reproduce the results, but there may be some variation because of sample variance or minor variations in their interpretation of the protocol or method.

**Reviewer Confidence:**

3: Pretty sure, but there's a chance I missed something. Although I have a good feel for this area in general, I did not carefully check the paper's details, e.g., the math, experimental design, or novelty.

---

> ### Author Rebuttal · Authors · 2023-08-29
>
> **Q1: HingeABL's improvement appears to be less significant when the base model is more powerful.**
>
> Thanks. The seemingly less significant improvement is not relevant to more powerful models, but it is a nature result of a higher baseline with improved loss functions. Among the three base models, ATLOP employs ATL loss, DocuNet employs Balanced Softmax loss, and KD-DocRE employs AFL loss. Both Balanced Softmax and AFL are the improvements of ATL. Therefore, replacing the better loss functions with HingeABL results in a smaller improvement.
>
> **Q2: Whether a strong base method could potentially alleviate the threshold problem?**
>
> We find that stronger base models do not necessarily mitigate the thresholding problem better.
> According to our experiment, KD-DocRE is stronger than DocuNet, but it has a higher number of FN_CRKs, and the good results may be due to better relation representation, better resolution of the relation type imbalance, etc. The optimization method they use does not primarily address the positive/negative imbalance.
> The statistics are shown in the table below.
>
> ||#FN_CRK|#FN_CRK with HingeABL|
> |---|---|---|
> |ATLOP|3273|1438|
> |DocuNet|1494|1418|
> |KD-DocRE|2688|1275|
>
> **Q3: Do you have further experiments to provide additional statistics on the FN-CRK in these three methods?**
>
> According to our experiment, no matter how strong or weak the underlying model is, replacing the loss function with HingeABL, the number of FN_CRKs all gets decreased for three base models (ATLOP, DocuNet, and KD-DocRE). The statistics are shown in the table in Q2.

---

### Official Review · Reviewer_yRY8 · 2023-08-04

**Soundness:** 3

**Excitement:**

3: Ambivalent: It has merits (e.g., it reports state-of-the-art results, the idea is nice), but there are key weaknesses (e.g., it describes incremental work), and it can significantly benefit from another round of revision. However, I won't object to accepting it if my co-reviewers champion it.

**Paper Topic And Main Contributions:**

The paper proposes the Adaptive Hinge Balance Loss to address the imbalance problem in document-level Relation Extraction (RE) tasks. In document-level RE, most entity pairs do not hold any relations, leading to a highly imbalanced class distribution. To model the difficulty of classification, the authors use the Separate Adaptive Thresholding (SAT), measuring the distance between a classification threshold score and the predicted score of each relation.
The proposed Adaptive Hinge Balance Loss puts the "Hinge weighting" on the loss function, drawing more focus on hard, misclassified relations. The weight is a nonlinear function based on SAT.
The experiment results on the Re-DocRED dataset demonstrate the superiority of this approach over other balancing methods. Analysis also shows this method indeed reduces false negatives and false negatives with correct ranking of classes.

**Reasons To Accept:**

1. The idea of using Separate Adaptive Thresholding combined with Hinge Weighting to modulate loss function is reasonable.
2. The results suggest this method is effective to some extent.


**Reasons To Reject:**

1. As the authors point out, with this method "as the false negative predictions decrease, the false positive predictions increase". So the improvement of the performance is limited, in terms of F1 scores.

**Reproducibility:**

4: Could mostly reproduce the results, but there may be some variation because of sample variance or minor variations in their interpretation of the protocol or method.

**Reviewer Confidence:**

4: Quite sure. I tried to check the important points carefully. It's unlikely, though conceivable, that I missed something that should affect my ratings.

---

> ### Author Rebuttal · Authors · 2023-08-29
>
> **Q1: With this method, as the false negative predictions decrease, the false positive predictions increase.**
>
> Thanks. This phenomenon does not only occur in our method, but in all the methods compared in the paper, as shown in the table below. For example, compared with ATL, AFL reduces #FN by 409, but increases #FP by 404. **All the methods have this limitation. HingeABL has the largest reduction (196) of total false predictions, i.e. #(FP+FN).**
>
> |Loss Function|F1|#FP|#FN|#(FP+FN)|
> |---|---|---|---|---|
> |ATL|73.29|2859|4081|6940|
> |Balanced Softmax|73.68|3299|4222|7521|
> |AFL|74.15|3263|3672|6935|
> |SAT|73.46|2910|4008|6918|
> |MeanSAT|74.68|5196|2488|7684|
> |**HingeABL**|**75.15**|3297|3447|**6744**|
>
> **Q2: The improvement of the performance is limited, in terms of F1 scores.**
>
> As shown in the table in Q1, HingeABL achieves a performance gain of 1.86 points in F1 score, which is **the most significant compared with other balancing methods**. Besides, the gain of 1.86 points in F1 score is already **a very large performance improvement in the field of document-level relation extraction**.
>
> In order to get a clear picture of how much performance improvement can be achieved by approaches in the field of document-level RE, here we list several widely implemented methods and their performance on the Re-DocRED dataset.
>
> |Method|F1|
> |---|---|
> |ATLOP (baseline)|77.56|
> |DocuNet|77.87 (+0.31)|
> |KD-DocRE|78.28 (+0.72)|
> |SSR-PU|78.86 (+1.30)|
> |ATLOP+HingeABL (Ours)|79.79 (**+2.23**)|
>
> For example, KD-DocRE is an effective and popular method, which is the improvement of ATLOP. **It only achieves a performance gain of 0.72 F1 compared with ATLOP**.
>
> In our paper, we observe that the "positive/negative class imbalance" problem accounts for the majority of prediction errors, which is ignored by previous methods. Based on the observation, we thoughtfully implement HingeABL through a combination of hinge loss and cross-entropy loss and **achieve a performance gain of 2.23 F1, which is significant and substantial**. HingeABL addresses the "positive/negative class imbalance" that has not been carefully considered by previous methods.

---

### Meta-Review · Area_Chair_V43d · 2023-09-20

**Recommendation:** 4

**Metareview:**

This paper focuses on the imbalance problem of N/A. The reviewers and the authors respectively give the detailed comments and responses. I suggest the authors fully consider the reviews and make precise revisions (especially the missing baseline).

---

### Decision · Program_Chairs · 2023-10-07

**Decision:**

Accept-Findings

**Comment:**

This paper focuses on the imbalance problem of N/A. The reviewers and the authors respectively give the detailed comments and responses. I suggest the authors fully consider the reviews and make precise revisions (especially the missing baseline).